

# Combined effects of boundary layer dynamics and atmospheric chemistry on aerosol composition during new particle formation periods

Liqing Hao[1], Olga Garmash[2], Mikael Ehn[2], Pasi Miettinen[1], Paola Massoli[3], Santtu Mikkonen[1], Tuija Jokinen[2], Pontus Roldin[4], Pasi Aalto[2], Taina Yli-Juuti[1], Jorma Joutsensaari[1], Tuukka Petäjä[2], Markku Kulmala[2], Kari E. J. Lehtinen[1,5], Douglas R. Worsnop[1,2,3], Annele Virtanen[1]

[1]Department of Applied Physics, University of Eastern Finland, Kuopio, Finland
[2]Department of Physics, University of Helsinki, P.O. 64, Finland
[3]Aerodyne Research Inc., Billerica, MA 08121-3976, USA
[4]Division of Nuclear Physics, Department of Physics, Lund University, P. O. Box 118, SE-221 00, Lund, Sweden
[5]Finnish Meteorological Institute, Kuopio, Finland

*Correspondence to:* Liqing Hao (hao.liqing@uef.fi) and Annele Virtanen (annele.virtanen@uef.fi)

**Abstract.** Characterizing aerosol chemical composition in response to meteorological changes and atmospheric chemistry is important to gain insights into new particle formation mechanisms. A BAECC (Biogenic Aerosols-Effects on Clouds and Climate) campaign was conducted during the spring 2014 at SMEAR II station (Station for Measuring Forest Ecosystem-Aerosol Relations) in Finland. The particles were characterized by a high-resolution time-of-flight aerosol mass spectrometer (HR-ToF-AMS). A PBL (planetary boundary layer) dilution model was developed to assist interpreting the measurement results. Right before nucleation events, the mass concentrations of organic and sulfate aerosol species were both decreased rapidly along with the growth of PBL heights. However, the mass fraction of sulfate aerosol of the total aerosol mass was increased, in contrast to a decrease for the organic mass fraction. Meanwhile, an increase of LVOOA (low-volatility oxygenated organic aerosol) mass fraction of the total organic mass was observed, in distinct comparison to a reduction of SVOOA (semi-volatile OOA) mass fraction. Our results demonstrate that, at the beginning of nucleation events, the observed sulfate aerosol mass was mainly driven by vertical turbulent mixing of sulfate-rich aerosols between the residual layer and the newly formed boundary layer, while the condensation of sulfuric acid played a minor role in interpreting the measured sulfate mass concentration. For the measured organic aerosols, their temporal profiles were mainly driven by dilution from PBL development, organic aerosol mixing in different boundary layers and/or condensation of organic vapors, but accurate measurements of organic vapor concentrations and characterization on the spatial aerosol chemical composition are required. In general, the observed aerosol particles by AMS are subjected to joint effects of PBL dilution, atmospheric chemistry and aerosol mixing in different boundary layers. During aerosol growth periods in the night time, the mass concentrations of organic aerosols and organic nitrate aerosols were both increased. The increase of SVOOA mass correlated well with the calculated increase of condensed HOMs (highly oxygenated organic molecules) mass. To our knowledge, our





results are the first atmospheric observations showing a connection between increase in SVOOA and condensed HOMs during the night time.

## 1. Introduction

Atmospheric aerosols have significant impacts on global climate change via direct and indirect forcing, air quality and human health (IPCC 2013). Accurate quantification of their sources and atmospheric evolution are necessary to reduce the uncertainties of global climate predictions.

New particle formation has been recognized as a significant aerosol source in the atmosphere (e.g. Nieminen et al., 2018; Gordon et al., 2016). Their subsequent growth is an important source of cloud condensation nuclei (CCN) relevant to climate change (Kerminen et al. 2012). Numerous measurements have shown that new particle formation takes place in the planetary boundary layer (PBL) on a global scale (Zhang et al., 2012; Kulmala, M. and Kerminen, V.-M., 2008; Kulmala et al., 2004). The PBL is the lowest layer of the troposphere, which is characterized by strong turbulent motions. The PBL is affected by the strength of the solar radiation (Stull, 1988). During the daytime, solar radiation heats the planet surface and induces convective turbulent motions and a well-mixed PBL develops. During the nighttime, several sub-layers are present when the planet surface cools down: the uppermost part is defined as a residual layer where emissions and background aerosol from the previous day are accumulated, and the part close to the ground develops to be a stable nocturnal boundary layer because solar heating ceases and the ground cools by emissions of infrared radiation, leading to increasing atmospheric temperature with height. This layer suppresses turbulence and vertical mixing. The height of PBL is an important variable in atmospheric science as it controls the vertical profiles of mixing ratios of gases and particles in the atmosphere. Formation of new aerosol particles depends greatly on concentrations of certain gas-phase species and their subsequent transformation (e. g. Tröstl et al., 2016; Ehn et al., 2014; Kulmala et al., 2000). These gas species can oxidize or undergo atmospheric reactions and transform to vapors with vapor pressures low enough to nucleate or condense. When the gas species are emitted into the planetary boundary layer, the dynamics of PBL affect the spatial distributions of the aerosol precursor species and thus their reaction products. Consequently, an influence of PBL dynamics on the new particle formation potential could be possible (Schobesberger et al., 2013; Wehner et al., 2010; O'Dowd et al., 2009; Laakso et al., 2007; Nilsson et al., 2001). For example, O'Dowd et al., (2009) found that the most intensive nucleation occurred just above the canopy. In contrast, nucleation rates were also enhanced in the upper PBL (Schobesberger et al., 2013; Wehner et al., 2010; Nilsson et al., 2001). The different nucleation studies above implied the effects of boundary layer developments on the nucleation events or/and different nucleation mechanisms taking place at different altitudes of boundary layer. To better understand the new particle formation mechanisms inside the boundary layer, the characterization of aerosol chemical composition and concentrations along with PBL development are needed.

This work presents the results of characterization of the atmospheric aerosols in accordance with the PBL development using high-resolution time-of-flight aerosol mass spectrometry (HR-Tof-AMS) in a boreal forest environment in Finland. We explored the impacts of PBL dynamics and atmospheric chemistry on the observed aerosol chemical composition and



mass concentrations before and during the nucleation events. Even though the minimum size cutoff for AMS is 35nm in a vacuum aerodynamic diameter (Zhang et al., 2004), the aerosol particles measured by AMS are dominated by the size range of Aitken- and accumulation modes, we have observed distinct variations of aerosol chemical composition during new particle formation periods. Hence, this study provides useful information to advance the understanding of new particle
formation mechanisms.

## 2. Experimental

### 2.1 Measurement site

During BAECC campaign (Petäjä et al., 2016), an aerosol mass spectrometer was deployed to measure the particle mass concentration, chemical composition and size distribution at SMEAR II ground station (Station for Measuring Forest
Ecosystem-Aerosol Relations) in Hyytiälä forestlands in Southern Finland (61° 51'N, 24° 17'E) during 8th April—20th June, 2014 (Hari and Kulmala, 2005). The site is located on a hill (180 m ASL) surrounded by boreal forest, mainly consisting of Scots pine, Norway spruce, birch and aspen. The populated city of Tampere lies approximately 50 km southwest to the site.

### 2.2 AMS operation and data processing

During the campaign, the real-time measurements of aerosol particles were performed using an Aerodyne soot particle
aerosol mass spectrometer (SP-AMS) (Onasch et al., 2012). The SP-AMS is a standard Aerodyne high resolution time-of-flight AMS equipped with an intracavity laser vaporizer (1064nm), in addition to the tungsten vaporizer used in a standard AMS (Canagaratna et al., 2007, DeCarlo et al., 2006). During the campaign, SP-AMS was operated at 5-min saving cycles alternatively switching between EI (electron ionization)-mode and SP-mode. In EI-mode, only the tungsten vaporizer was used to measure non-refractory chemical species such as organics, sulfate, nitrate, ammonium, and chloride. In SP-mode,
AMS was operated with both intracavity laser vaporizer and the standard tungsten vaporizer to produce mass spectra of laser-light absorbing particles such as refractory black carbon (BC) and non-refractory species. Standard mass-based calibrations were performed for the ionization efficiency (IE) by using mono-dispersed pure ammonium nitrate particles (Jayne et al., 2000). Regal black (REGAL 400R pigment black, Cabot Corp.) was used to determine the IE of BC, in a similar operation procedure as nitrate calibration.
The AMS data were processed using Tof-AMS Data Analysis Toolkit SQUIRREL version 1.57H and PIKA version 1.16H in Igor Pro software (version 6.22A, WaveMetrics Inc.). In addition, an improved-ambient elemental analysis was processed by using APES V1.06 (Canagaratna et al., 2015). For mass concentration calculations, the dataset from EI-mode was analyzed for reporting the non-refractory aerosol species and PMF simulations and the data in SP mode for the reported black carbon (BC). Default relative ionization efficiency (RIE) values of 1.1, 1.2, 1.3 and 1.4 were applied for nitrate,
sulfate, chloride and organics, respectively. The RIE for BC and ammonium were 0.11 and 2.65, respectively, as determined from the mass-based ionization efficiency calibration. After a comparison to the volume concentration from Differential





Mobility Particle Sizer (DMPS) measurement, a particle collection efficiency factor of 0.85 was applied to account for the particle losses in the aerodynamic transmission lens and vaporizer.

Further analysis was performed by applying Positive Matrix Factorization (PMF) technique on the high-resolution mass spectra (Paatero and Tapper 1994; Ulbrich et al. 2009). For the current study, the organic and error matrices in the m/z range 12-129 amu of high-resolution mass spectra were generated in PIKA in EI mode. The time series and errors of $NO^+$ and $NO_2^+$ ions were integrated into the organic and error matrices for PMF analysis. The combined organic and inorganic matrices were then fitted using the PMF evaluation tool. The PMF technique on the combined organic and inorganic matrix has been elaborately introduced in Sun et al. (2012) and Hao et al. (2014). The technique is capable of separating organic factors from inorganic ones and has been widely applied to quantify the particulate organic nitrate aerosols in recent studies (Xu et al., 2018; Zhang et al., 2016; Kortelainen et al., 2016; Xu et al., 2015; Hao et al., 2014, 2013). In this study, the PMF was evaluated with 1 to 10 factors and Fpeak from -1.0 to 1.0.

## 2.3 Gaseous compound concentrations

Highly oxygenated organic molecules (HOMs) were measured by a chemical ionization atmospheric pressure interface time-of-flight mass spectrometer (CI-APi-TOF-MS) (Jokinen et al., 2012). The CI-APi-TOF was run in negative ion mode with $NO_3^-$ acting as the reagent ion. In the campaign, the CI-APi-TOF signals were severely interfered by water clusters in the daytime. Hence, only a limited amount of data in the night time measurement were used and more details are provided in Sec. 3.4.

The molecular concentrations of sulfuric acid (SA, $H_2SO_4$) were estimated by a proxy approach (Mikkonen et al., 2011). The SA proxy was approximated by photo-oxidation reactions of sulfur dioxide ($SO_2$) under global radiation. The estimated SA concentrations are at an order of $10^6$ molecules/m³ (Fig.S1, Supplementary Information, SI), which is in the same magnitude as those observed in Hyytiälä region (Petäjä et al., 2009).

## 2.4 Planetary boundary layer

The height of planetary boundary layer (PBL) was provided by the GDAS simulation (Global Data Assimilation System) at the campaign site and they were validated against Radiosonde measurements. The GDAS data are comparable to the measurement (Fig.S2, SI) and thus were used to interpret the data.

## 2.5 Condensation sink (CS)

The CS for sulfuric acid was estimated by the approach developed by Pirjola et al., (1999) and was briefly expressed as the following equation (Lehtinen et al., 2003; Dal Maso et al., 2005):

$$CS = 2\pi D \int D_p \beta_m(D_p) n(D_p) dD_p = 2\pi D \sum \beta_m(D_{p,i}) D_{p,i} N_i \qquad (1)$$



where $D_{p,i}$ is the diameter of a particle and $N_i$ is the particle number concentration in a size bin i,. D represents the diffusion coefficient, and βm is the transitional correction factor. The estimation of CS was conducted from the size distribution measured by a Differential Mobility Particle Sizer (DMPS). Since DMPS measured the distribution of dry particles, the effect of ambient relative humidity on the hygroscopic growth of particles was also taken into account based on the parameterization of growth factors derived in Hyytiälä area (Laakso et al., 2004).

A similar approach was also employed for HOMs CS. Since there exists a large amount of different HOMs molecules, to simplify data processing, we virtually reconstructed a model molecule $C_{13}H_{21}O_9$ based on the α-pinene SOA chamber studies of Ehn et al. (2014) and used it to estimate the condensation sink of HOMs in this study. The molecule represents an average structure for HOMs with a molar weight of 321 g/mol, close to 325 g/mol for the average molar mass of the HOMs produced from α-pinene photochemistry. In the calculation, a total diffusion volume for HOMs of 310.2 was applied from the estimation of the atomic diffusion volumes of 15.9 for C, 6.11 for O and 2.31 for H (Reid et al., 1987). The final CS for HOMs is approximately half of that for SA (Fig.S3, SI).

## 2.6 Supporting measurements

The aerosol number concentration and size distribution in a size range of 3-1000 nm were measured by a DMPS. Other supporting measurements included $O_3$ (ozone), $SO_2$, CO (carbon monoxide), $NO_x$ (nitrogen oxides) and meteorological parameters (wind speed, wind direction, precipitation, temperature, solar radiation and relative humidity etc.) that are recorded continuously at the site throughout the year. The data are available by downloading in SmartSMEAR (https://avaa.tdata.fi/web/smart/, Junninen et al., 2009).

## 3. Results and discussion

### 3.1 Concentration and chemical composition of aerosols during nucleation events

The mass concentration of individual chemical species and chemical composition of PM1 particles varied greatly during the campaign period (Fig. 1). Generally, the total aerosol volume concentration derived from non-refractory species together with refractory BC measured by AMS correlated well with the collocated measurement by DMPS ($r^2$=0.93 and slope=1.07), assuming a density of 1.75 g cm$^{-3}$ for ammonium nitrate, ammonium sulfate and ammonium bisulfate, 1.52 g cm$^{-3}$ for ammonium chloride, 1.3 g cm$^{-3}$ for organics and 1.77 g cm$^{-3}$ for BC (Salcedo et al., 2006) (Fig. S4). The total PM1 mass concentrations varied between 0.14 μg m$^{-3}$ and 26.3 μg m$^{-3}$, with an average mass concentration of 3.1 ± 3.2 μg m$^{-3}$ (mean ± standard deviation). Individual species mass concentrations also varied considerably, especially the organic component, showing a dependence on temperature. A similar observation of increased organic aerosol concentrations with rising ambient temperature has also been reported at the same site (Corrigan et al., 2013). Averaged over the campaign (Fig. 1C), organic component accounted for 67.6 %, sulfate for 17.7 %, ammonium for 6.8 %, nitrate for 2.2 %, black carbon for 5.6 % and chloride for < 1 % of the total PM1 mass.





In this campaign several new particle formation episodes were observed. We focused on four events that were not perturbed by significant air mass changes, characterized by low variation of wind direction (±30°) and low wind speed (below 2 m s$^{-1}$). To demonstrate an overall picture of the evolution of the total aerosol chemical composition during the different phases of new particle formation events, we have divided each event into two stages, which we call "before" and "during" the nucleation and growth. The time period 'before' here is related to the 'nominal' nucleation starting time that was defined as the point when the DMPS started to see the nucleated aerosols up to a size of 3 nm in diameter (refer to Fig. S5, SI). As can be seen in Fig. 2, the particle number and mass concentrations decrease during the morning hours before the nucleation starts. At the same time the mass fraction of sulfate increases and organics decreases. The decrease in number and mass concentration is due to the increasing PBL height associated with sunrise. On average, the aerosol mass concentration was 0.9 μg m$^{-3}$ during the four events, which was approximated as one-third of the average campaign value. Concerning the chemical composition, the aerosol mass was comprised of 62.3% organic, 20.4% sulfate and 5.9% ammonium during events (Fig. 1D), in contrast to the values of 67.6 %, 17.7%, and 6.8%, for campaign average (Fig.1C), respectively. The difference in aerosol composition between the nucleation and other times was caused by combined effects of PBL development and atmospheric chemistry and more discussion will be provided next.

## 3.2 Aerosol components by positive matrix factorization

To gain insight into the relative variation of individual components, a PMF analysis of the high-resolution organic mass spectra together with $NO^+$ and $NO_2^+$ ions was conducted. After a detailed evaluation of mass spectral profiles, time series and comparison to the results of formerly reported mass spectra and supporting measurement data from other instruments, a 5-factor solution at Fpeak=0 separated four organic factors and one inorganic factor, and thus was chosen. A 4-factor solution did not extract out the inorganic nitrate factor and thus missed one meaning factor compared to a 5-factor solution. A 6-factor solution split the factor 2 of 5-factor solution to two sub-factors and did not produce more meaningful factors and thus was abandoned. More diagnostics plots are provided in Sec. S1 in the SI.

The mass spectra profiles and time series of the five factors are shown in Fig. S10. The organic component was resolved into one LVOOA (low volatility oxygenated organic aerosol) factors and three SVOOA (semi-volatile OOA) factors. The factors SVOOA1, SVOOA2 and SVOOA3 were merged to generate a new factor by means of a mass-weighted combination, representing a combined less oxygenated organic factor. As a result, an improved 3-factor solution is reported in this paper (refer to Fig. S12). A detailed description of the PMF results are elucidated in Sec. S3 in the SI. Generally, the average mass concentrations of LVOOA and SVOOA were $0.52 \pm 0.51$ μg m$^{-3}$ and $1.48 \pm 2.22$ μg m$^{-3}$, accounting for 15.3 % and 43.6 % of the total aerosol mass, respectively.

The determination of particulate organic nitrate by PMF is presented in Sec. S4 in the SI. The average mass concentration of organic nitrates was $0.01 \pm 0.02$ μg m$^{-3}$. The organic nitrate aerosols play a role in the growth stage of newly formed particles and more discussion is provided in Sec. 3.4.



### 3.3 Observations of new particle formation events

The aerosol size distributions, time series of aerosol components, and meteorological parameters during the four new particle formation events are depicted in Fig. 2 and Fig. S14. Around the onset of nucleation, we have systematically observed several features concerning the aerosol composition (in the orange shaded area): (1) the mass concentrations of organics and sulfate both decreased (Panel C); (2) the mass fractions of sulfate aerosol to the total aerosol mass increased, in distinct contrast to the decrease of organic mass fractions (panel D); (3) the mass concentrations of both LVOOA and SVOOA decreased (panel F), but the mass fractions of LVOOA to the total organic concentration increased while SVOOA decreased (panel E).

The rapid decrease of aerosol concentrations coincided with the increasing PBL height when the sunrise started to heat the surface (Fig. 3). Hence, we developed a PBL dilution model to assist interpreting the measurement results. In the model, we hypothesize that the observed decrease in aerosol mass is caused by the increased dilution due to an increasing PBL height. In the calculation, we converted the heights of boundary layer to dilution factors as a function of time (Eq.(2)). The starting time was defined as the point when the PBL height started to develop and the ending time when the PBL height maximum was reached. The development of PBL caused a dilution effect on the aerosol concentrations, e.g. in Fig. 3C, the mass concentration of aerosol species was decreased by 94% at the end. Meanwhile, in the model we did not include the downwards turbulent motions of aerosol particles from the residual layer to boundary surface layer. Instead, this factor was considered and examined separately and results are shown below.

$$Dilution\ factor = {PBL_{\_to}}\big/{PBL_{\_t}} \qquad (2)$$

where $PBL_{\_to}$ is the PBL height when the PBL starts to develop, and $PBL_{\_t}$ represents the height of PBL at time $t$.

The time series of the measured and modeled organic and sulfate mass concentrations are presented in Fig. 4. The modeled mass concentrations were calculated based on the above-mentioned model, assuming that the change in concentrations was controlled only by the dilution caused by increasing PBL height, i.e. by dividing the initial concentration (before the sunrise) by the dilution factor (diamond markers in Fig. 4). We see that the general trends in the calculated curves follow the measurements for both organic and sulfate species. During the night, the aerosol particles are concentrated in the shallow boundary layer. In the morning, the solar radiation initiates an increase in the PBL height, leading to dilution and decreasing concentrations.

However, for sulfate species, we observed much higher measured concentrations than the modeled ones (top panels, Fig. 4). The modeled sulfate concentrations can account for only $20.5 \pm 4.2$ % of the measurement results at the end of studied periods, suggesting that other factors also contributed to the measured sulfate species. Since meteorological conditions seem to be stagnant during those periods (relatively stable wind speed (WS) and wind direction (WD) in panel I, Fig. 2 and Fig S14), the perturbation of advected sulfate species from local sources likely had negligible effect on their concentrations. Thus, there are only two alternative explanations for the discrepancy between the measured and modeled concentrations: (1) photochemistry driven formation of sulfuric acid (SA) took place in the gas phase and the condensation of sulfuric acid onto



the preexisting particle contributed to the observed aerosol mass concentrations; and/or (2) sulfate-rich particles were downwards transported from the residual layer to newly developed mixed boundary layer when the PBL developed right after sunrise.

To estimate if the condensation of SA could explain the measured sulfate aerosol mass during the studied time periods, we estimated the condensed SA concentration from the condensation sink and the SA proxy:

$$mass_{SA} = \int_0^t CS_{SA} \cdot [SA] \qquad (3)$$

where $mass_{SA}$ refers to the accumulated SA mass from gas to particle phase over time period $t$. CS is the condensation sink term for SA and $[SA]$ represents the steady-state concentration of gas phase SA, which was approximated by the SA proxy approach of Mikkonen et al. (2011).

The calculated sulfate concentrations after taking into account the effects of both the PBL dilution and the SA condensation are shown as the dashed lines in Fig. 4 (in upper panels). For sulfate aerosols, taking into account both the dilution and condensation of SA, the calculation results account for $28.2 \pm 5.0$ % of measured sulfate concentrations. Furthermore, according to our calculations, in order to explain the discrepancy between the measurement and the pure dilution model by only SA condensation, it would require an $H_2SO_4$ concentration of $0.4{\sim}2.8{\times}10^9$ molecules·cm$^{-3}$ (based on Eq. (3)), which is at least two orders of magnitude higher than the $H_2SO_4$ concentrations typically found in Hyytiälä (Petäjä 2009). Thus, it is highly likely that the rest of uncounted sulfate (71.8 %) was originated from vertical turbulent mixing of the sulfate-rich aerosols from the residual layer into the newly formed boundary layer, while the condensation of sulfuric acid played a minor role in contributing to the uncounted sulfate mass concentrations. Our results are consistent with the study by Morgan et al. (2009) showing that the vertical structure of sulfate aerosol profile was primarily driven by PBL dynamical processes. The results also highlight the fact that measurements on the planetary surface are not always representative of aerosol properties at elevated altitudes. To understand the atmospheric aerosol properties, climate impacts, and to make accurate aerosol model predictions, a simultaneous representation of the aerosol vertical distribution is necessary.

For organic aerosols, the dilution model (solid green lines, Fig. 4) can relatively well track the measurements (green diamond markers), reproducing $61.8 \pm 36.7$ % of the measurement results at the end (upper panels, Fig. 4). Further analysis of the organic aerosol component by PMF reveals that the mass concentrations of LVOOA and SVOOA both decreased. The mass fractions of LVOOA to the total organic mass, however, were increased while the SVOOA mass fractions were decreased during the morning hours when the new particle formation took place (Panels E and F, Fig. 2). In addition, when taking into account the PBL dilution in calculated LVOOA concentrations (pink lines in Figure 4) the calculated concentrations are clearly lower than the measured concentrations while calculated SVOOA concentrations were generally higher than the measured ones. The discrepancy between the modeling and measurements could be interpreted by the mixing of LVOOA rich aerosol from the residual layer to the ground layer, and/or by the partitioning of organic vapors between aerosol and gas phase. To get more detailed and quantitative information on these processes, gas phase measurements of



organic vapors would be needed. Unfortunately, quantitative data of organic vapor concentrations is not available for this measurement campaign. Overall, it is likely that the temporal profiles of organic concentrations in the beginning of new particle formation were subjected to the interplay of mixing of LVOOA rich OA from residual to boundary layer and partitioning of organic vapors during the boundary layer evolution and new particle formation.

## 3.4 Aerosol growth

The growth of atmospheric aerosols from nucleation mode size range to the size at which they can be activated to cloud condensation nuclei (CCN) is an important step in linking new particle formation to climate change. To understand what is behind the growth mechanisms is thereby of scientific significance. In this study, we also investigated the changes in aerosol composition during the growth period of newly formed particles by AMS (the periods are indicated by grey bars in Figure 2 and S14). To simplify the analysis, we selected the growth periods in relatively stagnant meteorological conditions with steady WS and WD (in the gray bars, Figs. 2 and S14). In addition, the PBL was fully developed during the studied growth periods. We observed several interesting phenomena in the evolution of aerosol composition at the stage of aerosol growth. Firstly, we observed an obvious increase of organic aerosol mass concentration, comparing to the relatively stable concentration of sulfate aerosol (Panel C). Second, PMF results further revealed that the mass concentration of SVOOA increased. In contrast, the mass of LVOOA was almost unchanged (Panel F). Third, the mass concentration of organic nitrate was also enhanced (Panel G,).  Last, the growth of aerosol particles took place in the nighttime in all four events.

To investigate in more detail the contribution of highly oxygenated organic molecules to the growth during the night time and their contribution to the increased mass concentration of SVOOA, we used the nitrate CI-APi-TOF data. As mentioned before, due to the instrumental problem (CI-APi-TOF measurement were interfered by water signal), the absolute concentrations of measured HOMs were not quantitative. Hence, instead of using absolute signals, we used normalized ion signals to qualitatively interpret our results. In addition, we chose the ten most abundant ions observed during the aerosol growth periods, which were least affected by the water interference. The ten ions were monomers, dimers and nitrogen-containing ions including $C_{10}H_{14}O_7$, $C_{10}H_{16}O_7$, $C_{10}H_{14}O_9$, $C_{10}H_{16}O_9$, $C_{10}H_{14}O_{10}$, $C_{10}H_{16}O_{10}$, $C_{10}H_{14}O_{11}$, $C_{20}H_{32}O_{10}$, $C_{20}H_{32}O_{11}$ and $C_{20}H_{31}O_{13}N$. In our calculations we assume that all measured HOMs have extremely low vapor pressure and that we can use the condensation sink approach to calculate the mass of condensed HOMs during the night time growth (Eq. 4).

$$mass_{HOMs} = \int_0^t CS_{HOMs} \cdot [HOMs] \qquad (4)$$

where $mass_{HOMs}$ refers to the accumulated amount from gas to particle phase over time period $t$ for HOMs. CS is the condensation sink for HOMs. The [HOMs] represents the steady-state concentrations of HOMs measured by CI-APi-TOF.

Fig. 5 shows the increase in mass concentration of LVOOA and SVOOA as a function of relative increase in mass due to the HOM condensation during the night time aerosol growth periods (gray bars in Fig. 2). An excellent linear correlation was established between the condensed amounts of organic vapor and the increased mass of measured particles during the particle growth periods: the correlation coefficients r are in the range of 0.87-0.98. In contrast, the condensed HOMs show





negative correlation with LVOOA, with r between -0.17 and -0.60 (top panels, Fig. 5). The results demonstrate that the night time condensation of HOMs correlate well with SVOOA mass increase indicating that the HOMs contribute to the mass increase of SVOOA and total organic mass during the night time. Earlier laboratory and field studies have linked HOM to SOA formation (Ehn et al., 2014), but to our knowledge, our results are the first atmospheric observations showing a connection between increase in SVOOA and condensed HOMs during the night time.

As a constituent of organic aerosol, organic nitrate species were estimated to contribute $2.2 \pm 0.8$ % of the increase of aerosol mass at the growth stage. Note that this study measured only the nitrate functionality ($-ONO_2$) of organic nitrates. Taking into account the molar mass of $R-NO_3$ (where R represents organic components and we assume a lower limit of molar mass of 200 g mol$^{-1}$) (Xu et al., 2015), the fractional contribution of organic nitrate molecules ($RONO_2$) to the increased aerosol mass should be tripled (which should at least reach 6.6%). The results demonstrate that organic nitrates are a significant constituent of organic aerosol mass and are consistent with the studies showing that the organic nitrates are important for the increase of monoterpene SOA mass in chamber studies (Berkemeier et al., 2016). The results also suggest that ozonolysis and nitrate radical chemistry have crucial roles in contributing to the increased organic mass of atmospheric aerosols in the nighttime in boreal environment.

## 4. Conclusions

The measurements were conducted to characterize atmospheric submicron aerosol particles during new particle formation events in a boreal forest environment, Finland. The main goal was to investigate the temporal variation of aerosol species in response to the meteorological variation and atmospheric chemistry during new particle formation periods.

The aerosol composition during four focused nucleation events was dominated by organics (62.3%) and sulfate (20.4%) in terms of mass concentration. In the beginning of nucleation events, the mass concentrations of organic and sulfate aerosol components were both controlled by the boundary layer development. The temporal variation of sulfate aerosol mass concentration was mainly driven by the mixing of sulfate-rich aerosols from the above residual layer down to the newly formed mixed boundary layer in the first few hours after sunrise, while the condensation of sulfuric acid formed by photochemistry played a minor role. Based on our observations, we also hypothesize that the temporal evolution of organic concentration in the beginning of nucleation was caused by the interplay of mixing of organic aerosol from the above residual layer down to the boundary layer and/or possible condensation of organic vapors. During the night time, we observed an increase in the organic aerosol and organic nitrate mass concentrations, compared to the relative stable sulfate mass concentrations. The night time increase in organic mass was driven by the SVOOA components and the increase of SVOOA mass correlated well with the calculated increase of condensed HOM mass, indicating that the HOMs contribute to the mass increase of SVOOA and total organic mass during the night time.

**Acknowledgements**





The authors acknowledge Dr. Pekka Rantala for providing PTR-MS data, Dr. Antti Manninen and Mr. Kimmo Korhonen for measuring and analyzing PBL data. We also thank Dr. Aki Kortelainen, Dr. Hao Wang and Dr. Aki Pajunoja for maintaining AMS in the campaign. The financial supports by European Research Council (starting grants 355478 and 638703), the Academy of Finland Centre of Excellence program (decision no. 307331), Academy of Finland (259005), and UEF Postdoc
Research Foundation (930275) are gratefully acknowledged.

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



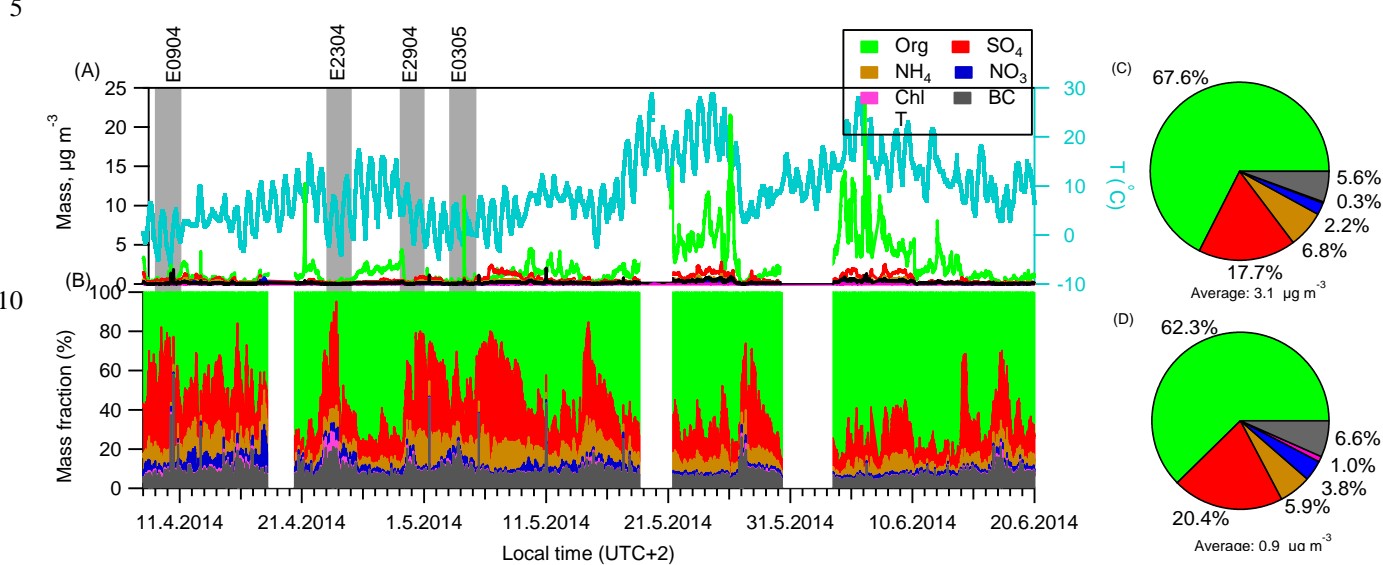

**Figure 1:** Time series and chemical composition of aerosol species determined by AMS in the campaign. The grey bars mark the four new particle formation events that this study focuses on. (A): Ambient temperature (T) and time series of aerosol mass concentrations. (B): Mass fractions of each aerosol component to the total aerosol mass concentrations. (C): Right pie charts show the average chemical composition for the campaign periods and (D): for the four new particle

20 formation periods.

25



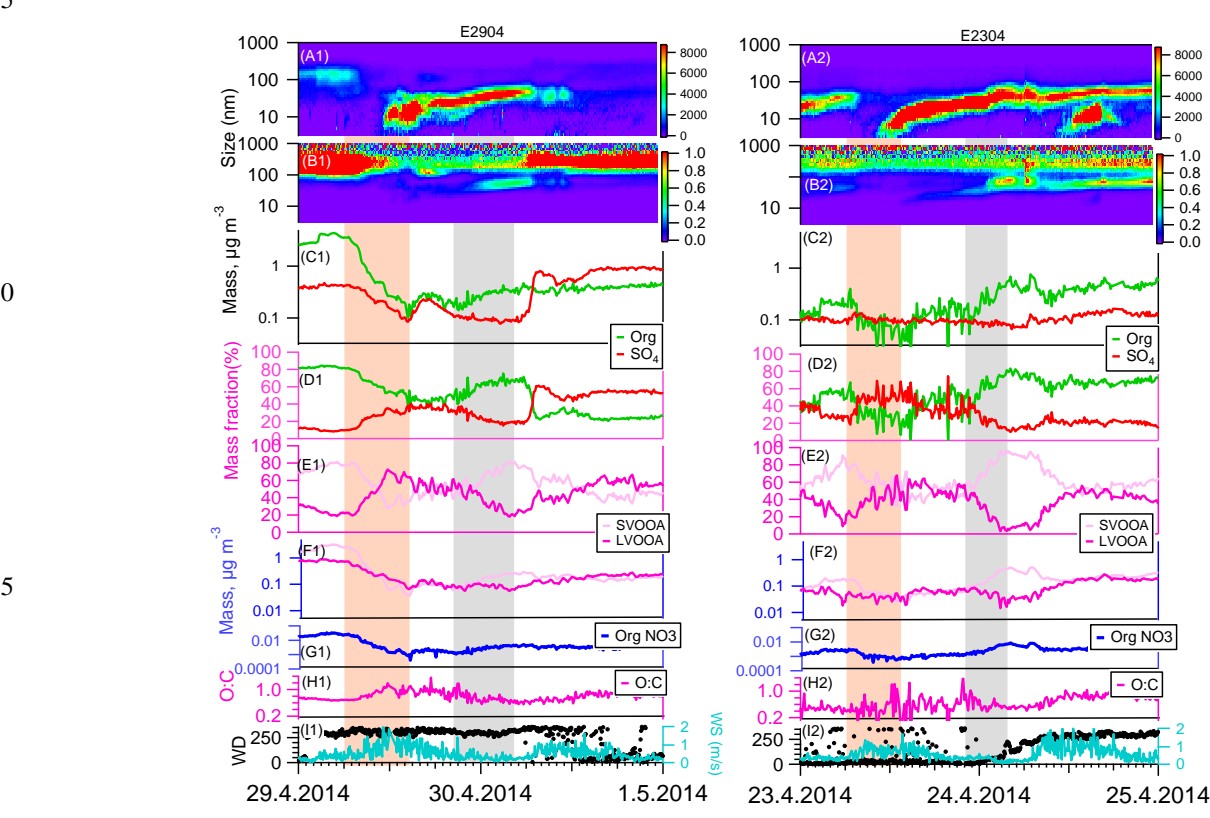

**Figure 2:** Observation of new particle formation events starting on 29.4.2014(E2904) and 23.4.2014 (E2304). The orange bars were marked for the analysis before and right after nucleation and the gray bars were for aerosol growth periods. Panels: (A) aerosol number size distributions from DMPS; (B) volume size distributions from DMPS; (C) mass concentrations of organic (green) and sulfate (red) species by AMS; (D) mass fractions of organic and sulfate to total aerosol mass concentrations; (E) mass fractions of LVOOA (pink) and SVOOA (light pink) to total organic aerosol mass concentrations; (F) mass concentrations of LVOOA and SVOOA species determined by PMF; (G) the time series of organic nitrate aerosol; (H) O:C ratio of organic species by AMS; (I) wind speed (WS) and direction (WD).





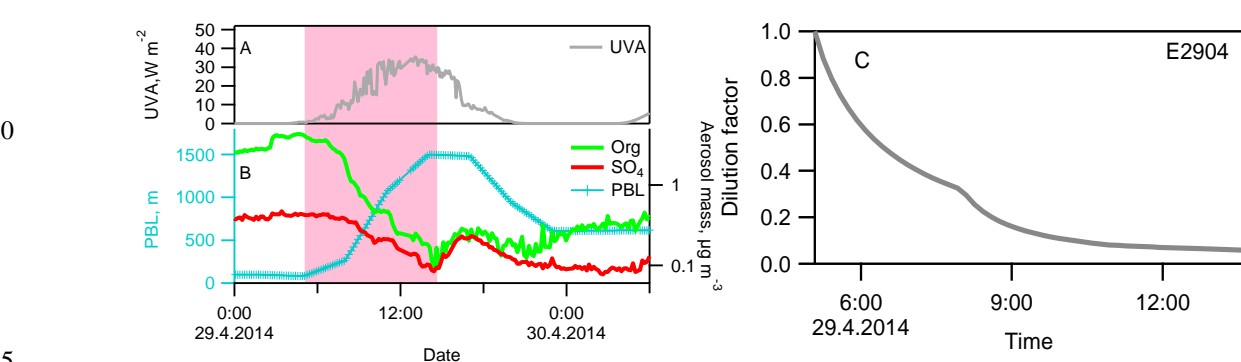

**Figure 3:** A closer view of the decrease of aerosol mass concentrations coinciding with PBL height development in E2904 (29.4.2014). (A) Ultraviolet (UV)-A radiation forcing; (B) the height of PBL and the mass concentrations of organic and $SO_4$ components measured by AMS; (C) dilution factor, estimated from PBL heights.

25

30





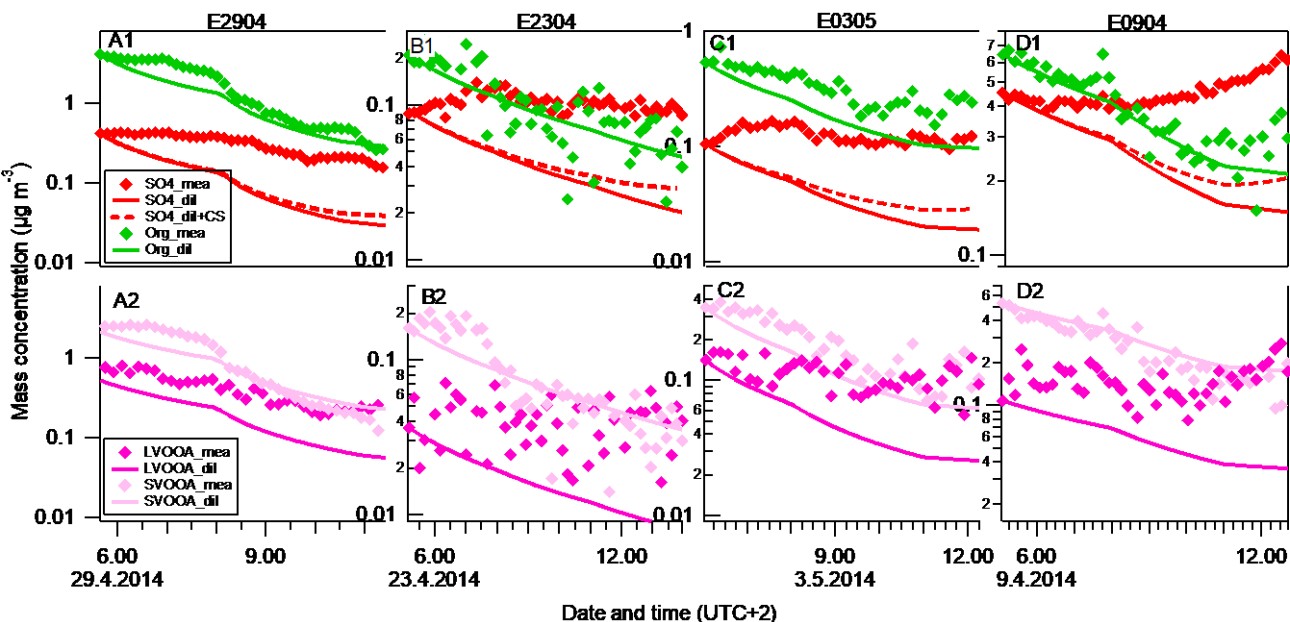

**Figure 4:** The time series of sulfate and organics (top panels), and LVOOA and SVOOA (bottom panels) aerosols depending on the dilution and vapor condensation effects before nucleation events in four events (orange bars in Figure 2). Diamond markers: the measured aerosol concentrations. Solid lines: the modelled aerosol concentrations assuming PBL heights to be an only controlling factor on aerosol mass concentrations. The aerosol concentrations were calculated by accounting for the air volume varying with PBL height. Dashed lines: the modelled aerosol concentrations accounted by both the dilution effect and condensation of sulfuric acid.





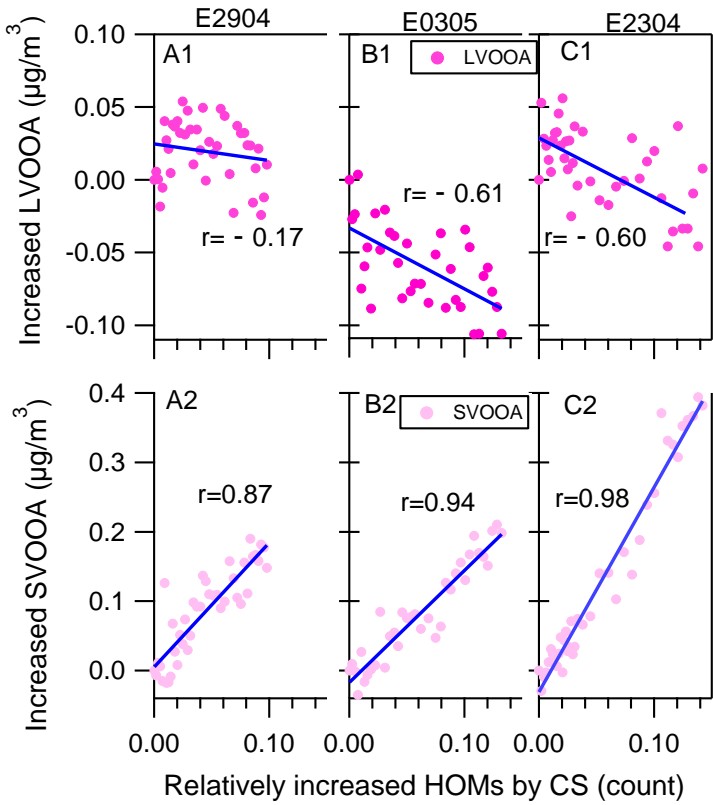

**Figure 5:** Comparisons of the increased mass of LVOOA (top panels) and SVOOA (bottom panels) in measurements (vertical axis) to the accumulated amount of HOMs condensed onto the particle phase (horizontal axis) at the particle growth stage (gray bars in Fig. 2). Note that AMS didn't work in E0904 growth period (event day of 9.4.2014) and thus the results are not shown.