# Peer review of "Combined effects of boundary layer dynamics and atmospheric chemistry on aerosol composition during new particle formation periods"

_Atmospheric Chemistry and Physics, 2018_

## Referee Comment (RC1) · Anonymous Referee #1 · 4 Sep 2018

The manuscript by Hao et al. investigated the effects of boundary layer dynamics and chemistry on aerosol composition during new particle formation (NPF) periods in Finland. The authors found that sulfate showed a much slower decrease than organics as a result of rising boundary layer during before the NPF period, which was likely due to the mix of residual layer with new boundary layer. During the growth period, they found the simultaneous increases in semi-volatile OOA and highly oxygenated organic molecules, suggesting a link between them. This manuscript is overall well written and I recommend it for publication after addressing the following comments.

Major comments:

[Figure]

1. The interpretation of slower decrease in sulfate than organics can have a third reason. As the boundary layer rises, organics can be diluted more because of evaporative loss associated with gas-particle partitioning, while the non-volatile sulfate does not. This points to another question of the PBL model.

2. The PBL model only considers the vertical dilution. In fact, as shown in Figure 2, the wind speed also has a significant change before the new particle formation period, increasing from ~0.3 m/s to 1.5 m/s. Although the wind speed is overall low, such an increase can have a big impact on horizontal dilution.

3. The quantification of AMS is a bit strange. The author used a collection efficiency of 0.85 according to the comparisons with DMPS measurements. However, AMS reported 9% higher mass concentrations than those of DMPS. It seems that the AMS CE was underestimated. Typical values of 0.5 or 1 are used in field campaigns. In addition, which size ranges of DMPS the authors use for comparisons, 3 – 1000 nm?

4. The authors identified three SV-OOA factors with different time series. I am wondering if HOMs correlates one of them or the combined one?

5. This manuscript missed the very important size-resolved chemical information from SP-AMS.

---

## Short Comment (SC1) · 4 Oct 2018

This is an interesting study, which aims at linking aerosol chemical composition during particle formation events to local meteorology and chemical factors. In section 3.3, the authors discuss how dilution due to PBL growth, mixing in of aerosol from the residual layer and aerosol formation by partitioning of organic vapors can affect the diurnal cycle of organic aerosol. These issues have been addressed quantitatively in a previous study on PBL effects on SOA concentrations at the same site by me and my co-authors (Janssen et al., 2012). Using a boundary layer-chemistry model in combination with observations, our study reached some conclusions which may be

relevant for the discussion in this study as well.

We found for instance that the local formation of SOA can play a role in the observed diurnal evolution of organic aerosol, but that for typical emissions at the SMEAR II station, its role is minor compared to that of PBL dilution and entrainment (figs. 8 and 9). Further, we tested the sensitivity of the simulated OA diurnal cycle to OA concentrations in the residual layer (Fig. 6). These different concentrations could be helpful in explaining the difference in dilution between the SV-OOA and the LV-OOA factors. For the LV-OOA, it seems likely that concentrations in the residual layer are relatively high, since it could have been formed elsewhere and have been advected over large distances. I hope these findings are useful for a more detailed discussion on the diurnal cycle of OA.

Best regards,

Ruud Janssen

Reference

Janssen, R. H. H., Vilà-Guerau de Arellano, J., Ganzeveld, L. N., Kabat, P., Jimenez, J. L., Farmer, D. K., van Heerwaarden, C. C. and Mammarella, I.: Combined effects of surface conditions, boundary layer dynamics and chemistry on diurnal SOA evolution, Atmos Chem Phys, 12(15), 6827–6843, doi:10.5194/acp-12-6827-2012, 2012.

---

## Referee Comment (RC2) · Anonymous Referee #2 · 16 Oct 2018

The authors present analysis of SP-AMS data combined with modelled data on the boundary layer height and supporting data of aerosol size distribution measurements. Their man finding is that sulfate aerosol mass concentration behaves differently than the organic aerosol during NPF event times: the sulfate concentration decreases less than the organic aerosol, and the same type of behaviour can be found for LVOOA with respect to SVOOA.

There are two main explanations for this observation: either it is a) a mixing effect of the PBL and the residual layer, with two different types of aerosols mixing, or b) there is formation of sulphate/LVOOA aerosol during the same period.

[Figure]

The authors study explanation b) by exploring the possibility of low- or nonvolatile vapors condensing on the existing aerosol and comparing the resulting aerosol mass to the observed one. Based on the discrepancy of the thus computed aerosol mass and the observations, the authors then conclude that the condensation of sulphuric acid cannot explain the sulphate time evolution and the sulphate aerosol is entrained from the residual layer. Similar results and conclusions are obtained for LVOOA aerosol.

This result is an interesting and important one, and the paper is well written and very well suited for ACP. I suggest that the paper is published, but I think that the conclusions could be supported more strongly by some additional discussion of the the possible origin an composition of the entrained aerosol and the mixing process:

Firstly, hat is the origin of the sulphate aerosol in the residual layer? Can its composition be estimated? (even approximately, e.g. amount/fraction of sulphate, LVOOA, SVOOA in the residual layer could be back-calculated from the data quite easily, I think).

Secondly, is there any indication of an external mixture when the mixing is occurring? In the case of a mixing, I guess there would be two different modes of aerosol (the entrained, sulphate-dominated mode and the PBL mode with less sulphate), while in the case of sulphuric acid condensing, the sulphate should be evenly distributed over the pre-existing aerosol. Is there any data available to test this assumption (e.g. the size information in the SP-AMS)?

Below are also some more minor comments:

P7 l22: If the DF is defined as in Eq. 2, then the initial concentration should be multiplied and not divided by the DF. (Typo?)

P8 l6: It seems that the mass_SA depends strongly on the CS_SA; if I understand correctly, the SA concentration estimate from the proxy is also depending on CS_SA, and therefore the uncertainty in Eq. (3) is proportional to CS^2. I think if would be interesting to compute the uncertainty in this, and also indicate the range of uncertainty

[Figure]

**[ACPD](about:blank)**

Interactive
comment

in the figures.

P9, l26: as it was just mentioned that [HOM] measurements were not quantitative, how was the concentration determined here?

p10, line 3: as the calculation is done assuming a non-volatile HOM, but then the statement says that HOM contributes to semi-volatile aerosol; could this be elaborated on as this seems contradictory?

---

## Author Comment (AC1) · 23 Nov 2018

**Responses to Referee 1:**

The authors thank the reviewer for her/his constructive and helpful comments. We have modified our manuscript according to the reviewer's comments and changes of the text and figures are marked in red in this response letter and in the manuscript.

*The manuscript by Hao et al. investigated the effects of boundary layer dynamics and chemistry on aerosol composition during new particle formation (NPF) periods in Finland. The authors found that sulfate showed a much slower decrease than organics as a result of rising boundary layer during before the NPF period, which was likely due to the mix of residual layer with new boundary layer. During the growth period, they found the simultaneous increases in semi-volatile OOA and highly oxygenated organic molecules, suggesting a link between them. This manuscript is overall well written and I recommend it for publication after addressing the following comments.*

We thank the referee for her/his positive comment on our manuscript.

*1. The interpretation of slower decrease in sulfate than organics can have a third reason. As the boundary layer rises, organics can be diluted more because of evaporative loss associated with gas-particle partitioning, while the non-volatile sulfate does not.*
Answer:
We thank the referee for the insightful comment. We do agree that the evaporation of organics could have taken place during the PBL development. In Fig. 4 in the manuscript, we have shown that the calculated SVOOA concentrations (light pink) were generally higher than the measured ones. As pointed out by the referee, this might be also due to SVOOA evaporation. Thus, we have rewritten the text from L28 in Page 8 to L6 in Page 9 by adding discussion on the evaporation issue.

We changed "In addition, when taking into account the PBL dilution in calculated LVOOA concentrations (pink lines in Figure 4) the calculated concentrations are clearly lower than the measured concentrations while calculated SVOOA concentrations were generally higher than the measured ones. The discrepancy between the modeling and measurements could be interpreted by the mixing of LVOOA rich aerosol from the residual layer to the ground layer, and/or by the partitioning of organic vapors between aerosol and gas phase. To get more detailed and quantitative information on these processes, gas phase measurements of organic vapors would be needed. Unfortunately, quantitative data of organic vapor concentrations is not available for this measurement campaign. Overall, it is likely that the temporal profiles of organic concentrations in the beginning of new particle formation were subjected to the interplay of mixing of LVOOA rich OA from residual to boundary layer and partitioning of organic vapors during the boundary layer evolution and new particle formation. " to "In addition, when taking into account the PBL dilution, in calculated LVOOA concentrations (pink lines in Figure 4) are clearly lower than the measured concentrations. The discrepancy between the modeling and measurements could be interpreted by the mixing of LVOOA rich aerosol from the residual layer to the ground layer, and/or by the partitioning of organic vapors between aerosol and gas phase. To get more detailed and quantitative information on these processes, gas phase measurements of organic vapors would be needed. Unfortunately, quantitative data of organic vapor concentrations is not available for this measurement campaign. For SVOOA component, the calculated concentrations were generally higher than the measured ones, which indicates that evaporation of SVOOA could have taken place due to the dilution mixing. Overall, it is likely that the temporal profiles of organic concentrations in this study were subjected to the interplay of mixing of LVOOA rich OA from residual to boundary layer and partitioning of organic vapors during the boundary layer evolution and new particle formation. "

*2. This points to another question of the PBL model. The PBL model only considers the vertical dilution. In fact, as shown in Figure 2, the wind speed also has a significant change before the new particle formation period, increasing from _0.3 m/s to 1.5 m/s. Although the wind speed is overall low, such an increase can have a big impact on horizontal dilution.*

Answer:

The wind speed (WS) was increasing in Fig. 2, with an average value of 0.72±0.44 m/s during the studied period in E0429. As pointed out by the reviewer, we didn't take the dilution due to the horizontal mixing in the account. As the Hyytiälä measurement station is surrounded by the large area of boreal forest, it can be assumed that the mixing is significantly less important than in the case of a point source. Anyhow, we would like to point out, if the horizontal mixing would have been taken into account in the calculations, the resulted dilution model concentrations would have been lower than those presented in Figure 4, further supporting the conclusion according to which the major part of sulfate is originated from vertical turbulent mixing of the sulfate-rich aerosols from the residual layer.

Accordingly, we deleted our argument in Line 29-32 in Page 7: "".

We also added a discussion on possible effects of horizontal dilution in Line 18, Page 8 in the manuscript: "It should be taken into account, that the horizontal dilution due to the increasing wind speed during the investigated period (wind speed increases from 0.3 to 1.5 m/s) was not taken into account in the analysis. Horizontal dilution could affect the dilution although in Hyyiälä, where the station is surrounded by the homogeneous forested area, the effect is minor."

*3. The quantification of AMS is a bit strange. The author used a collection efficiency of 0.85 according to the comparisons with DMPS measurements. However, AMS reported 9% higher mass concentrations than those of DMPS. It seems that the AMS CE was underestimated. Typical values of 0.5 or 1 are used in field campaigns. In addition, which size ranges of DMPS the authors use for comparisons, 3 – 1000 nm?*

Answer:

The estimation of CE was based on the comparison of AMS volume concentration to DMPS measurement. However, the comparison was prioritized to the new particle formation periods that this piece of work focuses on. Fig. C1 below shows a comparison of AMS-derived volume concentration (vol. conc.). vs DMPS-vol. conc. in E0429. AMS-vol. conc. was slighter higher than DMPS vol. conc. in the nucleation period (in orange bar), but was becoming lower in the aerosol growth period (in gray bar). The reasons for the slight inconsistency of two sets of data are (1) uncertainties of applied aerosol bulk densities when we convert AMS-mass concentration to volume concentration. This study applied a constant density of 1.75 $g/cm^3$ for the $SO_4$, $NO_3$ and $NH_4$ fragments. When the aerosol molecular composition varies in the atmosphere (for example, different amount of $(NH_4)_2SO_4$, $NH_4HSO_4$, $NH_4NO_3$ and $H_2SO_4$ in the orange bar from those in the gray shaded area), the approach could bring errors to the conversion from mass to volume concentrations, (2) possible differences in particle composition and hence phase related properties during these periods, and (3) possibly other minor issues with measurement fluctuation of two instruments. Hence, the employed of CE 0.85 is a compromised value after comparison between AMS- and DMSP-derived volume concentrations. For the entire campaign, AMS observed about 7% higher volume concertation than those of DMPS (not 9%, Fig. S4 in the supplementary materials). We still believe that CE 0.85 is within the range of uncertainties and is a reasonable estimation in this study.

The measurement size range of DMPS is 3-1000nm.

[Figure]

Figure C1. AMS- and DMPS- derived volume concentrations in E0429, assuming a CE of 0.85 for AMS. The orange and gray shaded bars were marked for analysis, the same periods indicated in Fig. 2 in the manuscript.

*3. The authors identified three SV-OOA factors with different time series. I am wondering if HOMs correlates one of them or the combined one?*

Answer:

The increase of SVOOA in the nighttime (gray bar in Fig. 2) is only due to the increase in SVOOA2 (see Fig.C2 below). Hence, HOMs correlates with SVOOA2. SVOOA2 shows good correlation with measured monoterpene VOCs and is identified as a biogenic SOA factor (Fig. S10 in supplementary information).

[Figure]

Figure C2. Time series of three SVOOA factors in the new particle formation periods.

*4. This manuscript missed the very important size-resolved chemical information from SP-AMS.*

Answer:

Unfortunately, the AMS signal was very low and hence also very noisy after the PBL dilution (total organic concentration app. 0.3 µg/m3) preventing us to analyze reliable changes in size-dependent composition before and after the PBL mixing (see Fig. C3). In addition, as it seems based on our results that the particle population entrained from residual layer is dominating the aerosol mass after the dilution, distinguishing the two population from AMS size resolved composition would require very good data quality. Hence, we didn't add the size information in the revised manuscript.

[Figure]

**Fig. C3** Size distributions of aerosol particles before and after PBL dilution mixing in four events. The time periods defined as before dilution and after dilution were tabulated in table S1. Panels (1)-(4): aerosol mass size distributions of Org and SO₄ before dilution by AMS; (2)-(8): aerosol mass size distributions of Org and SO₄ after dilution by AMS; (9)-(12): aerosol volume size distribution by DMPS. Note that AMS measures vacuum aerodynamic diameter and DMPS measures mobility diameter of aerosol particles.

Table S1 Time periods defined as before and after PBL dilution mixing.

| Events | Before dilution | After dilution |
|--------|-----------------|----------------|
| E2904  | 3:14-4:44       | 9:00-9:50      |
| E0305  | 4:03-5:58       | 9:34-12:00     |
| E2304  | 3:14-6:05       | 9:10-10:06     |
| E0904  | 2:05-4:20       | 9:57-11:42     |

---

## Author Comment (AC2) · 23 Nov 2018

**Responses to Referee 2:**

The authors thank the reviewer for her/his constructive and helpful comments. We have modified our manuscript according to the reviewer's comments and changes of the text and figures are marked in red in this response letter and in the manuscript.

*The authors present analysis of SP-AMS data combined with modelled data on the boundary layer height and supporting data of aerosol size distribution measurements. Their main finding is that sulfate aerosol mass concentration behaves differently than the organic aerosol during NPF event times: the sulfate concentration decreases less than the organic aerosol, and the same type of behaviour can be found for LVOOA with respect to SVOOA.*

*There are two main explanations for this observation: either it is a) a mixing effect of the PBL and the residual layer, with two different types of aerosols mixing, or b) there is formation of sulphate/LVOOA aerosol during the same period. The authors study explanation b) by exploring the possibility of low- or nonvolatile vapors condensing on the existing aerosol and comparing the resulting aerosol mass to the observed one. Based on the discrepancy of the thus computed aerosol mass and the observations, the authors then conclude that the condensation of sulphuric acid cannot explain the sulphate time evolution and the sulphate aerosol is entrained from the residual layer. Similar results and conclusions are obtained for LVOOA aerosol.*

*This result is an interesting and important one, and the paper is well written and very well suited for ACP. I suggest that the paper is published, but I think that the conclusions could be supported more strongly by some additional discussion of the possible origin and composition of the entrained aerosol and the mixing process:*

Answer: we thank the referee for the nice summary of the manuscript and positive recommendation for possible publication.

*1. Firstly, what is the origin of the sulphate aerosol in the residual layer? Can its composition be estimated? (even approximately, e.g. amount/fraction of sulphate, LVOOA, SVOOA in the residual layer could be back-calculated from the data quite easily, I think).*
Answer:
The origin of sulfate rich aerosol in the residual layer is likely related to the cloud processing of aerosols as a significant fraction of sulphate is formed in clouds (Ervens et al., 2011). It is also possible that sulphate rich aerosol has entrained from free troposphere. In addition, e.g. Sorooshian et al. (2010) and Hao et al. (2013) reported increased oxidation level of cloud residual particles suggesting that cloud processing of organics would lead to compounds having elevated O:C ratio. As suggested by the reviewer, we conducted the back-calculated approximation of aerosol chemical composition in the residual layer based on the comparison between our measurement and dilution modelling results. In the calculation we assumed that the partitioning of organic vapors is negligible, and the ammonium and nitrate were excluded in the analysis. On average, the aerosol mass in the residual layer was roughly comprised of $62.6 \pm 16.6\%$ $SO_4$, $35.6 \pm 15.4\%$ LVOOA and $1.8 \pm 7.1\%$ SVOOA in the four studied events, in a distinct contrast to aerosol chemical composition of $24.3 \pm 11.6\%$ $SO_4$, $17.2 \pm 0.1\%$ LVOOA and $58.5 \pm 0.1\%$ SVOOA in the stable surface boundary layer before PBL dilution was initiated.

We have added discussion in Page 9 that "Additionally, we conducted the back-calculated approximation of aerosol chemical composition in the residual layer based on the comparison between our measurement and dilution modelling results. In the calculation we assumed that the partitioning of organic vapors is negligible, and the ammonium and nitrate were excluded in the analysis. Hence, only Org and $SO_4$ aerosol were included in the analysis. On average, the approximated aerosol mass in the residual layer was comprised of $62.6 \pm 16.6\%$ $SO_4$, $35.6 \pm 15.4\%$ LVOOA and $1.8 \pm 7.1\%$ SVOOA in the four studied events, in a distinct contrast to aerosol chemical composition of $24.3 \pm 11.6\%$ $SO_4$, $17.2 \pm 0.1\%$ LVOOA and $58.5 \pm 0.1\%$ SVOOA in the stable surface boundary layer before PBL dilution

was initiated. The origin of sulfate rich aerosol in the residual layer is likely related to the cloud processing of aerosols as a significant fraction of sulphate is formed in clouds (Ervens et al., 2011). It is also possible that sulphate rich aerosol has entrained from free troposphere. In addition, e.g. Sorooshian et al. (2010) and Hao et al. (2013) reported increased oxidation level of cloud residual particles suggesting that cloud processing of organics would lead to compounds having elevated O:C ratio."

*2. Secondly, is there any indication of an external mixture when the mixing is occurring? In the case of a mixing, I guess there would be two different modes of aerosol (the entrained, sulphate-dominated mode and the PBL mode with less sulphate), while in the case of sulphuric acid condensing, the sulphate should be evenly distributed over the pre-existing aerosol. Is there any data available to test this assumption (e.g. the size information in the SP-AMS)?*

Answer:

This is a very interesting point. Unfortunately, the AMS signal was very low and hence also very noisy after the PBL dilution (total organic concentration app. 0.3 µg/m3) preventing us to analyze reliable changes in size-dependent composition before and after the PBL mixing (see Fig. C1). In addition, as it seems based on our results that the particle population entrained from residual layer is dominating the aerosol mass after the dilution, distinguishing the two population from AMS size resolved composition would require very good data quality. Hence, we didn't add the size information in the revised manuscript.

[Figure]

**Fig. C1** Size distributions of aerosol particles before and after PBL dilution mixing in four events. The time periods defined as before dilution and after dilution were tabulated in table S1. Panels (1)-(4): aerosol mass size distributions of Org and SO$_4$ before dilution by AMS; (2)-(8): aerosol mass size distributions of Org and SO$_4$ after dilution by AMS; (9)-(12): aerosol volume size distribution by DMPS. Note that AMS measures vacuum aerodynamic diameter and DMPS measures mobility diameter of aerosol particles.

Table S1 Time periods defined as before and after PBL dilution mixing.

| Events | Before dilution | After dilution |
|--------|-----------------|----------------|
| E2904  | 3:14-4:44       | 9:00-9:50      |
| E0305  | 4:03-5:58       | 9:34-12:00     |
| E2304  | 3:14-6:05       | 9:10-10:06     |
| E0904  | 2:05-4:20       | 9:57-11:42     |

Below are also some more minor comments:

*P7 l22: If the DF is defined as in Eq. 2, then the initial concentration should be multiplied and not divided by the DF. (Typo?)*

Answer:

The reviewer is correct. It was a typo in the manuscript and we have fixed it.

*P8 l6: It seems that the mass_SA depends strongly on the CS_SA; if I understand correctly, the SA concentration estimate from the proxy is also depending on CS_SA, and therefore the uncertainty in Eq. (3) is proportional to CS^2. I think if would be interesting to compute the uncertainty in this, and also indicate the range of uncertainty in the figures.*

Answer:

The estimation of SA proxy is a function of dry size CS and relative humidity $(CS_{dry} \times RH)^{-0.13}$, as indicated in Eq. 11 in Mikkonen et al (2012), whereas $CS_{SA}$ for mass_SA is corrected by RH. Thus, defining the uncertainty of condensated sulfuric acid in Eq. (3) is not as straightforward as the reviewer suggested.

To make estimate for uncertainty of condensated SA by applying equation of $\int_0^t CS \times SA$ in Eq.(3), we applied a formula from Farrance & Frenkel (2012):

$$y = x_1 \times x_2 \qquad\qquad (C1)$$

$$\left(\frac{u(y)}{y}\right)^2 = \left[\left(\frac{u(x_1)}{x_1}\right)^2 + \left(\frac{u(x_2)}{x_2}\right)^2\right] \qquad\qquad (C2)$$

where u (y) refers to uncertainty for variable y, u($x_1$) and u($x_2$) are uncertainties for variable x.

By applying Eqs (C1) and (C2) to our case, we obtain:

$$SD_{CS \times SA}^2 = \left[\left(\frac{SD_{CS}}{CS}\right)^2 + \left(\frac{SD_{SA}}{SA}\right)^2\right] \times (CS \times SA)^2 \qquad\qquad (C3)$$

where $SD_{CS}$ and $SD_{SA}$ represent the uncertainties for condention sink and sulfuric acid, $SD_{CS \times SA}$ is the uncertainty of multiplied CS by SA.

In the calculation, we estimated an uncertainty vale of 40% for SA and 20% for CS. The uncertainty for $CS \times SA$ is about 44.7% based on Eq. (C3). We took a 1.6 times of this value as an upper boundary and 0.4 times as a bottom boundary. The uncertainty range for condensated SA in Eq.(3) of the manuscript roughly is 17.9% -71.5%. However, because of the difficulties in estimating the SA proxy and CS uncertainties, we did not include these results in the manuscript.

*P9, l26: as it was just mentioned that [HOM] measurements were not quantitative, how was the concentration determined here?*

Answer:

It should be noted that x-axis of the Fig. 5 doesn't represent the quantitative HOM concentration, but rather the relative change in the HOM concentration during the growth period. Hence, we did not attempt to determine any absolute concentration. We have now brought this up now more clearly in the manuscript text.

We have added a sentence in Line 31, Page 9: "It should be noted that x-axis of the Fig. 5 doesn't represent the quantitative HOM concentration, but rather the relative change in the HOM concentration during the growth period."

We also changed the legend of x-axis in Fig. 5 to "Estimated cumulative HOM condensation (a.u.)".

*p10, line 3: as the calculation is done assuming a non-volatile HOM, but then the statement says that HOM contributes to semi-volatile aerosol; could this be elaborated on as this seems contradictory?*

Answer:

We thank the reviewer for the good comment. The volatility bins of AMS-PMF-derived LVOOA and SVOOA component vary a lot. For example, the PMF SVOOA consisted of 50% SVOA ($C^*$ =1-100 µg m$^{-3}$), 42% LVOA ($C^*$ =$10^{-1}$-$10^{-3}$ µg m$^{-3}$) and 6 % ELVOC ($C^* \leq 10^{-4}$ µg m$^{-3}$) in Paris (Paciga et al., 2016). A study conducted in Hyytiälä in 2014 showed a poor correlation of AMS PMF-SVOOA to VTDMA-derived SVOA ($r^2$=0.16) (Hong et al., 2017), which could indicate a wide range of volatility bins of PMF-SVOOA. Thus, it is still possible that HOMs contributes to both PMF-SVOOA and LVOOA.

References:
Ervens, B., Turpin, B. J., and Weber, R. J.: Secondary organic aerosol formation in cloud droplets and aqueous particles (aqSOA): a review of laboratory, field and model studies, Atmos. Chem. Phys., 11, 11069-11102, 2011.
Farrance, I. and Frenkel, R.: Uncertainty of Measurement: A review of the rules for calculating uncertainty components through functional relationships, The Clinical biochemist. Reviews, 33(2), 49-75, 2012.
Hao, L., Romakkaniemi, S., Kortelainen, A., Jaatinen, A., Portin, H., Miettinen, P., Komppula, M., Leskinen, A., Virtanen, A., Smith, J.N., Sueper, D., Worsnop, D.R., Lehtinen, K.E.J., and Laaksonen, A.: Aerosol Chemical Composition in Cloud Events by High Resolution Time-of-Flight Aerosol Mass Spectrometry. Environ. Sci. Technol., 47, 2645-2653, 2013.
Hong, J., Äijälä, M., Häme, S. A. K., Hao, L., Duplissy, J., Heikkinen, L. M., Nie, W., Mikkilä, J., Kulmala, M., Prisle, N. L., Virtanen, A., Ehn, M., Paasonen, P., Worsnop, D. R., Riipinen, I., Petäjä, T., and Kerminen, V.-M.: Estimates of the organic aerosol volatility in a boreal forest using two independent methods, Atmos. Chem. Phys., 17, 4387-4399, https://doi.org/10.5194/acp-17-4387-2017, 2017.
Mikkonen, S., Romakkaniemi, S., Smith, J. N., Korhonen, H., Petäjä, T., Plass-Duelmer, C., Boy, M., McMurry, P. H., Lehtinen, K. E. J., Joutsensaari, J., Hamed, A., Mauldin III, R. L., Birmili, W., Spindler, G., Arnold, F., Kulmala, M., and Laaksonen, A.: A statistical proxy for sulphuric acid concentration, Atmos. Chem. Phys., 11, 11319-11334, 2011.
Paciga, A., Karnezi, E., Kostenidou, E., Hildebrandt, L., Psichoudaki, M., Engelhart, G. J., Lee, B.-H., Crippa, M., Prévôt, A. S. H., Baltensperger, U., and Pandis, S. N.: Volatility of organic aerosol and its components in the megacity of Paris, Atmos. Chem. Phys., 16, 2013-2023, https://doi.org/10.5194/acp-16-2013-2016, 2016.
Sorooshian, A., Murphy, S. M., Hersey, S., Bahreini, R., Jonsson, H., Flagan, R. C., Seinfeld, J. H.: Constraining the contribution of organic acids and AMS m/z 44 to the organic aerosol budget: On the importance of meteorology, aerosol hygroscopicity, and region, Geophys. Res. Lett., 37, L21807, DOI: 10.1029/2010GL044951, 2010.

---

## Author Comment (AC3) · 23 Nov 2018

We would like to thank Dr Ruud Janssen for pointing us their earlier work on the combined effects PBL evolution and atmospheric chemistry on diurnal evolution of SOA. Their study using boundary layer –chemistry model and experimental observations is interesting and indisputably relevant to our study and their results support the findings of our study. We have now discussed and referred to the earlier work on page 9 of the modified manuscript.